# Exploring Iberian Peninsula Lamiaceae as Potential Therapeutic Approaches in Wound Healing

**DOI:** 10.3390/ph16030347

**Published:** 2023-02-24

**Authors:** Mário P. Marques, Laura Mendonça, Beatriz G. Neves, Carla Varela, Paulo Oliveira, Célia Cabral

**Affiliations:** 1Coimbra Institute for Clinical and Biomedical Research (iCBR), Clinic Academic Center of Coimbra (CACC), Faculty of Medicine, University of Coimbra, 3000-548 Coimbra, Portugal; 2Center for Innovative Biomedicine and Biotechnology (CIBB), University of Coimbra, 3000-548 Coimbra, Portugal; 3Centre for Functional Ecology, Department of Life Sciences, University of Coimbra, 3000-548 Coimbra, Portugal; 4Chemical Process Engineering and Forest Products (CIEPQPF), Faculty of Medicine, University of Coimbra, 3000-548 Coimbra, Portugal; 5Centro de Neurociências e Biologia Celular (CNC), Center for Innovative Biomedicine and Biotechnology (CIBB), 3000-548 Coimbra, Portugal

**Keywords:** skin, wound healing, Lamiaceae, Iberian Peninsula

## Abstract

Skin tissue has a crucial role in protecting the human body from external harmful agents, preventing wounds that frequently demand proper healing approaches. The ethnobotanical knowledge of specific regions with further investigation on their medicinal plants has been paramount to create new and effective therapeutical agents, including for dermatological purposes. This review attempts, for the first time, to investigate the traditional applications of Lamiaceae medicinal plants that are already used by local communities in the Iberian Peninsula in wound healing. Henceforward, Iberian ethnobotanical surveys were reviewed, and the information about the traditional wound healing practices of Lamiaceae was comprehensively summarized. Afterwards, the scientific validation of each Lamiaceae species was exhaustively checked. From this, eight out of twenty-nine Lamiaceae medicinal plants were highlighted by their wound-related pharmacological evidence and are in-depth presented in this review. We suggest that future studies should focus on the isolation and identification of the active molecules of these Lamiaceae, followed by robust clinical trials that may confirm the security and effectiveness of such natural-based approaches. This will in turn pave the way for more reliable wound healing treatments.

## 1. Introduction

The skin consists of three main layers: the epidermis as the outermost layer, followed by the dermis which assembles on the hypodermis [1]. It is considered the human body’s largest organ, covering and protecting it from several external harmful agents, such as chemical agents. It also reduces electrolytes loss, regulates evapotranspiration and body temperature, plus it acts as an immune defence against microorganisms [2].

Wounds are among the most common ailments that affect skin [3]. They imply tissue injuring, usually by external agents, causing severe damage to the epidermis and possibly to the underlying connective tissue [4]. The disruption of the normal function and structure of the human skin tends to lead to several types of wounds such as burns, contusions, ulcers and cuts [5,6]. Nevertheless, skin wounds can also arise from secondary complications of other diseases, such as cancer and diabetes [6,7]. Wounds are classified as acute or chronic depending on the synergy of factors that led to the injury, signalling inhibitors, type of skin wound and personal pre-existing conditions [4]. Therefore, it is expected that acute wounds undergo normal healing processes in just a few days depending on the injury’s severity, discarding the hypothetic growth of microorganisms [6]. In fact, the proliferation of common bacteria, such as *Staphylococcus aureus* and *Pseudomonas aeruginosa*, leads to healing delay and complications [5]. In turn, chronic non-healing wounds are more challenging clinical conditions, mostly associated with circulatory problems and diabetes, in which the normal healing process is often stopped by excessive neutrophil infiltration, uncontrolled inflammation, increased production of reactive oxygen species (ROS) and even necrosis. Chronic wounds affect approximately six million people worldwide, especially older people, with an estimated 3 billion dollars per year impact in health care [4,6,8].

Conventional chemical drugs used to treat skin injuries often present problematic side effects and reduced efficacy. As for plant-derived extracts and antioxidant bioactive compounds, such as plant polyphenols, they represent a more cost-effective, secure and faster healing approach [6,9]. In fact, the Western Pharmacopoeia comprises only 1 to 3% chemically synthesized drugs for treating wounds and skin ailments, while nearly 30% of herbal-based formulations have been highlighted for their beneficial effects in this context. Furthermore, medicinal plants have been suggested as an important source of therapeutic alternatives with minor side effects. It represents the primary healthcare solutions of almost 65% of people worldwide and 80% of the population in developing countries [9,10].

In this sense, the search for pharmacological evidence and scientific validation of traditional practices using medicinal plants for dermatological conditions has been the aim of numerous surveys in distinct geographical regions, such as Italy [11], Saudi Arabia [12], the Balkans [9], Morocco [6,13], and noteworthy the Iberian Peninsula [14,15]. In particular, the traditional use of the family Lamiaceae has been highlighted in several ethnobotanical assays carried out within the Iberian Peninsula [16,17,18,19,20]. Usually, Lamiaceae species are rich essential oil-bearing plants with a great diversity of phenolic compounds, polyphenols, iridoids, diterpenoids, triterpenoids, saponins and, in some restricted cases, pyridine and pyrrolidine alkaloids [21]. The presence of such compounds has been linked to a wide range of well-documented bioactivities, namely the antioxidant [7,22,23,24,25], anti-inflammatory [4,26,27,28,29] and especially antibacterial activity [22,23,25,30,31,32,33]. Altogether, this positively influences the wound healing potential of the Lamiaceae species.

In this review, we focused on the Iberian ethnobotanical practices with Lamiaceae species used in the treatment of wounds. Some background information about skin anatomy, physiology, and the wound healing process is provided. The main contribution of this review is related to the pharmacological evidence collected for such plants. Some aspects regarding major phytochemical bioactive agents, key *in vitro* and *in vivo* experiments, and clinical trials are herein summarized.

## 2. Methodology

A comprehensive analysis of ethnobotanical and ethnopharmacological surveys regarding traditional practices of medicinal plants used in the Iberian Peninsula for wound healing was made. Among studies published in a lifespan of twenty-two years, from 2000 to 2022, a total of twenty-nine Lamiaceae plant species have been cited due to their wound healing properties. Afterwards, the search for pharmacological evidence of each Lamiaceae plant species revealed that only eight have received proper scientific validation for this purpose. The search was performed in databases, such as ScienceDirect, Scopus, PubMed, Web of Science and Google Scholar. The following keywords were applied individually and/or in combination: wound healing, skin, wound, burn, cut, injury, ulcer, phytochemistry, medicinal plants, Lamiaceae, Iberian Peninsula, Portugal, and Spain.

## 3. Skin Anatomy and Physiology

The skin is the outermost surface and the largest organ of the human body, accounting for approximately 8–20% of total human weight and covering an area of 1.8 m^2^ [2,5,34]. From an inside-out perspective, the hypodermis, dermis, and epidermis are the three main layers that shape skin anatomy. From an embryological perspective, the epidermis and associated appendages such as nails and hair, arise from the ectoderm, while the dermis and the hypodermis are mesoderm-derived cell layers [1].

The epidermis consists of 0.1 mm thick stratified squamous epithelium, and it is a barrier with several defensive roles, protecting from harmful pathogens and irritant agents [2,34]. This layer does not have blood vessels being supplied from a vasculature that arises from bellow [2]. Herein, the main epidermal cell-types are keratinocytes, which account for almost 95% [35]. Keratinization, the process that leads to the differentiation of keratinocytes, starts at the inner basal layer of the epidermis where these cells are actively dividing. As older keratinocytes are projected out of the epidermis by the new ones, turning from a synthetic phase to a degrative one, the epidermis starts to be subdivided into four distinct sublayers: the *stratum basale* or *stratum germinativum*, *stratum spinosum*, *stratum granulosum* and *stratum corneum* [34,35]. An additional sublayer called *stratum lucidum* may be found between *stratum corneum* and *stratum granulosum* of the palms and soles’ epidermis. Besides keratinocytes, melanocytes, Merkel and Langerhans cells are also found in the epidermal layer, however, in a lower amount [1,34]. On the other hand, several appendages such as nails, hair, and three types of sweat glands (sebaceous, apocrine, and eccrine) are considered to originate in the epidermis, but they penetrate into the dermis already during embryo development [34,35].

Skin’s durability, strength, and flexibility arises from the dermis. It is divided in two distinct strata. Firstly, the papillary dermis, the most superficial, is mainly made of connective tissue, elastic fibres, and some collagen, besides the vasculature and nerves. Secondly, the innermost dermal layer known as reticular dermis, comprises fibroblasts, mast cells, nerve endings, lymphatics, and muscles, and it also contains thick collagen bundles and blood vessels. According to various stages of development, health, or disease state, other cell types may be present within the dermal layer. These include extravasated leucocytes, common during inflammation or infection, and histiocytes or reticulin cells [34].

Finally, the hypodermis or subcutaneous fat, highly vascularized and fundamentally made of adipocytes, is localized between the dermis and the underlying muscle, playing crucial roles such as thermoregulation, protection from mechanical injuries, and an energy reservoir. The thickness of this layer has particular anatomic and individual specificity, reflecting the nutritional behaviour of each person [1,34].

## 4. Physiological Process of Wound Healing

Skin wounds will start a well-orchestrated succession of events to re-establish skin’s function and mechanical integrity. Therefore, the conventional healing process consists of four time-dependent stages: hemostasis, inflammation, proliferation, and wound remodeling [5,6,36].

### 4.1. Hemostasis

Hemostasis, the initial phase of the wound healing process, begins immediately after any injury, and it is known to last from 1 to 3 h. The repair process initiates with vasoconstriction, followed by thrombocytes’ granules release with subsequent aggregation into a platelet plug which impairs the blood flow. This is only possible due to the release of several growth factors, cytokines, and low molecular weight substances from the serum of injured blood vessels and to the degranulating platelets [5,6,37]. As a result, the established blood clot incorporates cross-linked fibrin and extracellular matrix (ECM) proteins, such as fibronectin, vitronectin, and thrombospondin [38,39,40]. In addition to the formation of mature clots, this fibrin polymerization serves as a scaffold for infiltrating cells, such as leukocytes, keratinocytes, and fibroblasts, during the subsequent phases of healing [6,37,40]. In fact, inflammatory cells are attracted to the area by chemokines and cytokines released by platelets. This prompts an invasion of neutrophils, macrophages, and lymphocytes that will conduct to the subsequent stage of the healing process [5,37].

### 4.2. Inflammation

The inflammatory stage begins with the influx of neutrophils, macrophages, and lymphocytes to the site of injury, and this can last from 24 to 48 h, or even up to one week. This stage focuses on controlling the bleeding, preventing bacterial growth and proliferation, and removing any cell residues from the wound area [5,6,41]. Thus, after neutrophils’ arrival into the wound, the phagocytosis process starts. This process is immediately continued by the macrophages, which result from the differentiation of the monocytes present in the blood. Consequently, the by-products of neutrophil apoptosis and cell fragments are phagocytosed by macrophages [5,6,42]. Besides eliminating bacteria and cell residues from the wound site, macrophages also release several growth factors and other molecules that stimulate fibroplasia and angiogenesis [5,43]. Cytokines, such as interleukin IL-6, IL-1β, IL-10, matrix metalloproteinases (MMPs), and growth factors, including vascular endothelial growth factor (VEGF), insulin-like growth factor 1 (IGF-1), basic-fibroblast growth factor (b-FGF), epidermal growth factor (EGF), and transforming growth factor (TGF-β) are some of the examples [6,40,44,45]. Finally, lymphocytes appear in the wound site to help collagenase regulation that is crucial for collagen remodeling in the subsequent phases [6,42]. 

### 4.3. Proliferation

Once the wound is free from damaged cells and associated residues, the proliferative phase begins, and it may take 20 days in acute wounds. It involves different sub-phases such as re-epithelialization, angiogenesis, and granulation tissue formation [5,6,46,47]. During re-epithelialization, mesenchymal stem cells (MSCs) differentiate into keratinocytes, which then migrate and proliferate to cover the wound. Thus, a new epidermal layer forms on the surface from the wound edges through the keratinocytes and the epithelial cells [48,49,50]. Angiogenesis, neovascularization, or new blood vessels formation as it is also known, is the next phase and it is promoted by growth factors secreted formerly during the hemostasis phase, such as VEGF, platelet-derived growth factor (PDGF), and FGF [6,47,51].

### 4.4. Remodeling

Remodeling is the last phase of the wound healing process, and it involves wound cytokine production and wound contracture. It can last from 21 days to a year after the wound has been inflicted. Thus, in response to the cytokines and growth factors present in the wound site, fibroblasts produce MMPs that will act according to an orchestrated balance by either destroying the provisional ECM [52] or by synthesizing a new one composed of different types of collagens, proteoglycans, hyaluronic acid, glycosaminoglycans, fibrin, and fibronectin [36,53,54]. This process will strengthen and structure the new formed tissue. Through myofibroblasts’ differentiation, it will also enhance wound closure and ultimately lead to scar formation [5,6,55,56,57]. As the wound heals, fibroblasts and macrophages undergo apoptosis, that is responsible for reducing their presence at the wound site. Furthermore, capillary growth ceases and blood flow and metabolism are reduced [6,36].

## 5. Traditional Use of Lamiaceae in the Iberian Peninsula

The Iberian Peninsula (Portugal and Spain) is a vast territory in the western Mediterranean, separated from the rest of the European continent by the Pyrenees mountains. This affords an interesting and important flora in this region [58]. The Iberian territories comprise nearly 25% of the 25,000 vascular plant species in the Mediterranean basin (the third greatest hotspot of plant biodiversity in the world) [59]. Noteworthy, there are nearly 6500 vascular plant species in the Iberian Peninsula, of which 1328 are endemic [58,59].

Bearing in mind this peculiar phytogeographical context, Lamiaceae has been highlighted since it comprises many plants known for their medicinal application in this specific region [16,17,18,19,20]. Lamiaceae is a worldwide spread botanical family that contains approximately 5600 species, distributed over 186 different genera, although other authors argue that this number may rise to 7200 plants species. This heterogeneous group of plants is highly represented in the Mediterranean region, especially in the Iberian Peninsula where it is estimated to have approximately 271 Lamiaceae species from 37 distinct genera [21]. In fact, it is suggested that the wide availability of wild Lamiaceae to the Iberian people communities, their appealing flowers, and their use for pleasant essential oil production has prompted their massive use in folk medicine ever since [18,19]. Notwithstanding, the medicinal importance of such a group of plants is equally denoted in other circum-Mediterranean territories [6,13,21]. Therefore, in this work, we reviewed Iberian ethnobotanical surveys published between 2000 and 2022. The gathered information is summarized in Table 1, referring to the traditional use of Lamiaceae for wound healing found in six Portuguese surveys [17,60,61,62,63] and nine Spanish works [14,15,18,19,20,64,65,66,67]. A total of twenty-nine Lamiaceae plants were found to be applied topically in different forms and preparations for the treatment of skin wounds.

## 6. Pharmacological Wound Healing Evidence of Iberian Lamiaceae

After a thorough analysis of the pharmacological evidence from the twenty-nine Lamiaceae plant species presented in Table 1, only eight were found to have proper scientific validation (Figure 1). These were reviewed in detail with clinical trials presented in Table 2, as well as key findings of *in vivo* and *in vitro* studies summarized in Table 3. The chemical structures of the most relevant phytochemicals found within these eight species are illustrated in Figure 2.

### 6.1. Lavandula stoechas L.

#### 6.1.1. Ethnobotanical Uses and General Considerations

*L. stoechas* (Figure 1A) has a circum-Mediterranean distribution, and it is widely known for its applications in cosmetic, food, perfumery, and pharmaceutical industries [6]. This lavender is traditionally used as a carminative, antispasmodic, expectorant, anticonvulsant, analgesic, sedative, and diuretic [8]. It is also widely referred in ethnobotanical literature for the treatment of skin injuries and burns, either as antiseptic or healing/vulnerary of wounds [8,19].

#### 6.1.2. Phytochemical Background

Flavonoids, tannins, sterols, coumarins, mucilages, and triterpenoids such as oleanolic and ursolic acids, have been identified as major classes of compounds in its hydroethanolic extract [85]. Recently, Baali et al. [8] reported that the prominent compound found in *L. stoechas’* methanolic extract is rosmarinic acid, besides luteolin-7-*O*-glucoside, luteolin-7-*O*-glucuronide, salvianolic acid B, and quercetin-3-*O*-galactoside (Figure 2) [8]. On the other hand, and similarly to other aromatic plants of the Lamiaceae family, *L. stoechas’* essential oils may present certain chemical variability. Even though the essential oils from several locations around the Mediterranean basin are mainly characterized by the presence of monoterpene ketones, such as fenchone and camphor, or by the oxygenated monoterpene 1,8-cineole, besides other volatiles such as linalool, linalyl acetate, terpineol, terpinen-4-ol, α-pinene, viridiflorol, camphene, among others [74,85].

#### 6.1.3. Pharmacological Evidence

A clinical trial by Vakilian et al. [69] evaluated the efficacy of *L. stoechas*’ essential oil as a wound healing agent in episiotomy, a common perineal incision in obstetric and midwifery. This study included 120 randomized primiparous women divided in two main groups: patients treated with a sitz bath containing five to seven drops of the essential oil, and a control group of patients treated with povidone-iodine^®^. On the 10th day postpartum, the incision wounds were evaluated, and 25 patients treated with the sitz bath did not manifest any pain (*p* = 0.06), similarly to those treated only with povidone-iodine^®^, suggesting the efficacy of *L. stoechas* essential oil in pain relief. Furthermore, it reduced redness in the episiotomy area when compared to the control group (*p* < 0.001). Following this investigation, subsequent studies have been made to evaluate the utility of other lavender species essential oils on episiotomy healing [86,87,88].

Regarding the treatment of burn wounds, another clinical trial was performed for 14 days with 111 randomized patients with second-degree burns. This trial suggested that the incorporation of the essential oils of *L. stoechas* and *Pelargonium roseum* Ehrh. In a cream containing *Aloe vera* L. significantly reduced pain in patients with superficial second-degree skin burns, from day 0 to the 7th (*p* = 0.014) and the 14th (*p* = 0.05), when compared to the control group comprising patients treated with the standard drug silver sulfadiazine^®^ (SSD) 1% cream [70].

The wound healing effects of *L. stoechas* using *in vivo* excision wound models have also been evaluated [8,74]. From this perspective, using Wistar albino rats, the methanolic extracts of *L. stoechas* and *Mentha pulegium* L. were incorporated in ointments at two distinct concentrations (5 and 10%) [8]. During 18 consecutive days the extract-containing ointments were topically applied (0.5 g per rat) once a day until the complete re-epithelization of the excision wounds. A wound contraction of 93.1 ± 2.88% and 97.19 ± 1.06% was observed on the 18th day in rats treated with ointments containing the *L. stoechas* extract at 5 and 10%, respectively. Besides that, wound contraction with the lavender-containing ointment (10%) group was statistically higher (*p* < 0.001) in comparison with the positive control group, an allantoin-based pharmaceutical formulation. The herbal ointment containing *L. stoechas* extract was promising for re-epithelization and cell migration, a critical step in the wound healing proliferative phase. Furthermore, appreciable granulation tissue and higher collagen quantity, without inflammatory signs such as oedema or erythema, were other observed characteristics. The authors further hypothesize that phytochemical compounds such as hydroxycinnamic acids, flavanols, flavones, and flavonols may explain the wound healing activity observed [8]. In another recent study, a cream prepared with the essential oil of *L. stoechas* at 0.5% showed the highest effect on excision wound models compared to the reference Madecassol^®^, a registered therapeutic cream. On the 4th, 11th, and 16th day, wound contractions were, respectively, 26.4%, 78%, and 96.3% for the group of rats treated with this herbal formulation, compared to 8.5%, 64.1%, and 86.1% for the control group. The authors concluded that the *L. stoechas* cream induced a significant decrease in the epithelization period, wound area and scar thickness, along with a significant increase in wound contraction. Moreover, this treatment also resulted in decreased inflammatory parameters and a great rate of tissue perfusion and proliferation, as well as remodeling and re-epithelization [74].

The capacity of *L. stoechas* to induce fibroblast proliferation and consequent *in vitro* wound healing was also explored in another study using an aqueous extract. This revealed that the growth and migration of fibroblasts was promoted at 24, 48, and 72 h transducing in a wound closure of 21.3%, 27.4%, and 29.2%, respectively [73].

Before the scientific evidence around *L. stoechas*, new drug delivery systems have been designed. Therefore, Mahmoudi et al. [22] synthesized silver nanoparticles (AgNPs) with a reductant methanolic extract of *L. stoechas* [22]. AgNPs presented antioxidant properties and antibacterial potential against *S. aureus* and *P. aeruginosa* which are common wound infecting bacteria. As these nanoparticles exhibited biocompatibility at an effective and non-toxic concentration (62.5 μg/mL), authors suggest their application for wound healing [22]. It is worth mentioning that other investigations have been carried out to develop innovative strategies for drug delivery of essential oils from other *Lavandula* species [89].

### 6.2. Marrubium vulgare L.

#### 6.2.1. Ethnobotanical Uses and General Considerations

In traditional medicine, *M. vulgare* (Figure 1B) is mostly used to treat gastrointestinal and respiratory diseases [90], and also for several skin conditions, either in the Iberian Peninsula [20] or in other countries of the Mediterranean region [3,13]. Interestingly, the scientific-based wound healing activity of this Lamiaceae has received some attention in the Mediterranean basin [3,7,23].

#### 6.2.2. Phytochemical Background

*M. vulgare* is a poor essential oil-bearing plant with low extraction yields ranging between 0.03% and 0.06%. However, several monoterpenes have been identified in its essential oil, such as camphene, *p*-cymol, fenchone, limonene, α-pinene, sabinene, and α-terpinolene (Figure 2) [90]. Considering terpenoids, *M. vulgare* is enriched in several diterpenes such as the labdane-type diterpene marrubin. This is responsible for the plant’s bitter characteristic and it is also a chemotaxonomic marker for the genus *Marrubium* [7,90]. The triterpenoids lupeol and oleanolic acid, as well as phytosterols such as β-sitosterol, have been reported for this Lamiaceae species [90]. On the other hand, the flavonoid family in *M. vulgare* is mainly represented by flavones such as luteolin, ladanein, and apigenin, flavone derivatives such as apigenin-7-*O*-glucoside and luteolin-7-*O*-glucoside, and the flavonol derivatives quercetin-3-*O*-galactoside and rutin. For a long time, and since this plant is part of the Lamioideae subfamily, rosmarinic acid was thought to be absent. However, a few reports have shown its presence in M. vulgare extracts and caffeic, ferulic, and chlorogenic acids [90].

#### 6.2.3. Pharmacological Evidence

Recently, Mssilou et al. [3] assessed the activity of the hydroethanolic extract of *M. vulgare* to heal skin burns induced on the dorsal part of rats during a period of 21 days. The hydroethanolic extract of *Dittrichia viscosa* (L.) Greuter and the ointment-based combination with *M. vulgare* extract was equally investigated. According to the observations, the topical application of the ointment with *M. vulgare* extract, also in combination with *D. viscosa*, recorded remarkable and progressive wound closure at the 21st day, when compared to the controls that were unable to induce complete wound healing in the same period of time. Moreover, equal promising results were observed for inflammation and pain, other key parameters of the wound healing process [3]. Indeed, de Souza et al. [91] demonstrated the analgesic properties of an hydroalcoholic extract of *M. vulgare* regarding different models of pain in mice. In another study, Yahiaoui et al. [23] showed that the acetonic extract of *M. vulgare* improved the quality of the scar tissue and wound contraction by around 93.79% after 14 days, compared to control rats treated with Madecassol^®^ with a wound closure of around 96.55%.

Amri et al. [7] conducted an *in vitro* cell-based investigation with a methanolic extract. Authors found that in a non-toxic concentration (5 μg/mL), the extract showed to promote migration and proliferation of dermal fibroblasts (NHDF cell line), reaching complete confluence after 48 h of extract application. Besides that, the aqueous extract of *M. vulgare* demonstrated not only antioxidant potential, but also interesting hemostatic activity. The latter is suggested to arise from the presence of condensed tannins, which are well-known for their astringency activity. In fact, tannins are important hemostatic agents working positively in wound and burn healing. Moreover, they observed that wound healing is independent from the presence of the diterpene marrubiin [92].

### 6.3. Origanum vulgare L.

#### 6.3.1. Ethnobotanical Uses and General Considerations

*O. vulgare* (Figure 1C), commonly known as oregano, is a widespread herbaceous Lamiaceae found in Europe, North Africa, America, and Asia. A wide range of activities have been scientifically evidenced such as antimicrobial, antioxidant, anti-inflammatory, antitumor, antihyperglycemic, and anti-Alzheimer activities and in skin disorders [93]. Its ethnopharmacological uses are mainly related to respiratory conditions, such as cold symptoms, including cough, as well as digestive disorders and dermatological affections [4,6,93]. In the Catalonian region of Ripollès district, Spain, there are reports of using *O. vulgare*’s flowers in embrocation for their vulnerary properties [15].

#### 6.3.2. Phytochemical Background

*O. vulgare* is a rich essential oil-bearing plant and for this reason, the most relevant family of phytochemicals are the volatiles found in its essential oil. According to the analysis of several oregano essential oils, they are chemically polymorphic, existing with several chemotypes based on major compounds. Monoterpene phenolics such as carvacrol and thymol are prominent in oregano essential oils, along with linalool, γ-terpinene, *p*-cymene, and the sesquiterpenes β-caryophyllene and germacrene D. Besides essential oils, *O. vulgare* is also a rich source of flavonoids, tannins, and phenolic glycosides. From this point of view, luteolin-*O*-glucuronide and luteolin-7-*O*-glucoside have been pointed out as main flavonoid derivatives found in hydroalcoholic extracts, decoctions, and infusions. Smaller molecules such as caffeic, protocatechuic, vanillic, and *o*-coumaric acids have been equally identified, with rosmarinic acid as the major phenolic acid (Figure 2) [93].

#### 6.3.3. Pharmacological Evidence

In a randomized pilot petrolatum-controlled clinical trial, an ointment containing an *O. vulgare* aqueous extract was tested by topical application, twice a day, on patients with excision wounds. The group treated with *O. vulgare* ointment presented a significant improvement in comparison to the petrolatum-treated group regarding skin pigmentation, vascularization, thickness, relief, and pliability. It is worth mentioning that the ointment proved to be a good antimicrobial product on post-surgical excision injuries, namely against *S. aureus* [71].

The design of innovative drug delivery systems has been pursued by some authors [30,75,76]. In this sense, considering *in vivo* excision wound models, a study on synthesized titanium dioxide nanoparticles (TiO_2_.NPs) loaded with an oregano aqueous extract was made. Rats treated with TiO_2_.NPs presented a wound closure of 94% on the 12th day of evaluation while the control group remained at 86%. In addition, animals under this treatment showed an increased collagen content and degree of re-epithelization, as well better fibroblasts and macrophages aggregation [75]. Recently, a biocompatible pharmaceutical formulation with potential wound dressing was designed to overcome the instability and skin irritancy that may arise from the direct application of essential oils. In this study, the essential oil of *O. vulgare* was encapsulated in a poly (L-lactide-co-caprolactone) (PLCL)/silk fibroin (SF) nanofiber membrane through electrospinning. Results from the *in vivo* assays showed that the designed therapeutic system improved re-epithelialization and the formation of granulation tissue, and it also stimulated angiogenesis and collagen accumulation [30]. Afterwards, *in vivo* diabetic wound models were studied by the same team where the co-delivery nanofibrous membranes loaded with two bioactives, the essential oil of *O. vulgare* and zinc oxide, were tested. The bioactive multifunctional nanofibrous wound dressing system was revealed to promote tissue regeneration, re-epithelialization and collagen accumulation. Besides that, and according to the observed expression of VEGF, the angiogenic response was highly stimulated as well. The observations highlighted that the typical inflammatory process in diabetic wounds was also inhibited [76].

On the other hand, according to an *in vitro* scratch-based assay, the application of *O. vulgare* essential oil (25 μg/mL) promoted cell migration after 72 h [26]. Additionally, investigators stimulated keratinocytes with IFN-γ and histamine to induce ROS generation. They found that ROS levels were significantly reduced, along with the levels of inflammation and of the matrix metalloproteinase biomarkers [26]. Similarly, in another work, pre-inflamed human dermal fibroblasts were treated with *O. vulgare* essential oil, showing to inhibit both inflammatory and tissue remodeling biomarkers. In the end, authors suggested that the tested carvacrol-rich essential oil is a potential ingredient for skin-related products with anti-inflammatory activity [28]. Interestingly, carvacrol and thymol, phenolic monoterpenes abundant in the essential oils of plants from *Origanum* sp. and *Thymus* sp. genera, have demonstrated beneficial wound healing properties. According to Costa et al. [94], these monoterpenes act in the three distinct phases of the wound healing process: reducing inflammation, excessive ROS production, and infection, followed by the enhancement of angiogenesis, re-epithelialization, and tissue remodeling with final collagen synthesis along with the proliferation of dermal fibroblasts and epidermal keratinocytes.

### 6.4. Prunella vulgaris L.

#### 6.4.1. Ethnobotanical Uses and General Considerations

According to its ethnopharmacological uses, *P. vulgaris* (Figure 1D) is used around the world to treat several skin conditions [27]. In the Iberian Peninsula, its traditional topical application is mostly associated with its antiseptic activity, with flowers used in infusions or decoctions for washes and baths [15,18,63]. Vulnerary and healing properties are equally attributed to this plant species, being topically applied as a cataplasm or in baths for wounds healing [15,60].

#### 6.4.2. Phytochemical Background

Triterpenoids is the major and most important phytochemical group in *P. vulgaris*. They are skeletons of 30 atoms of carbon divided in three main types, oleanane, ursane, and lupane, and they may be present in the free, ester, or glycosylated form. In *P. vulgaris*, oleanolic acid and ursolic acid are pointed as the main triterpenoids (Figure 2) [95,96]. Steroids are also present and are mainly represented by phytosterols and their derived saponins, such as sitosterol and stigmasterol [96]. Other sterols have also been identified, such as daucosterol and α-spinasterol [95]. Besides triterpenes and sterols, a wide range of flavonoids have been equally identified, such as homoorientin, wogonin, quercetin-3-*O*-β-D-rhamnoside, kaempferol-3-*O*-β-D-glucoside, hesperidin, and acacetin-7-*O*-β-D-glucopyranoside [95]. Considering phenolic-derived compounds, *P. vulgaris* is rich in coumarins such as umbelliferone, scopoletin, and esculetin, and in phenolic acids such as rosmarinic, caffeic, and ellagic acids, the most important given the several associated pharmacological activities [78,95,96].

#### 6.4.3. Pharmacological Evidence

Recently, the wound healing activity of the *P. vulgaris* was assessed in an *in vivo* bioactivity-guided fractionation assay from its methanolic extract. Extracts at 1% were incorporated in ointment formulations and topically applied on wound models (incision and circular excision). Findings showed that, in the incisional wound, the ethyl acetate extract increased 39.3% of the tensile strength of the wound, while in the excisional wound a wound contraction of around 86.3% was observed after 12 days. Six bioactive compounds (ethyl rosmarinate, methyl arjunolate, ursolic acid, chlorogenic acid, rosmarinic acid, and methyl-3-epimaslinate) were identified in the ethyl acetate sub-extract proving to be the most effective for wound healing. Furthermore, ursolic, chlorogenic, and rosmarinic acids were shown to positively influence the anti-inflammatory and wound healing effects of *P. vulgaris* [27].

The thermal-induced wound healing properties of *P. vulgaris* were investigated in aqueous extracts. The *in vivo* study was performed for 14 days, and the plant-derived extract (10%) was incorporated in a cream, showing a better healing capacity for burn wounds when compared to the standard SSD^®^ cream. According to these findings, the topical application of *P. vulgaris* stimulated collagen production and antioxidant efficiency by lipid peroxidation suppression, a decrease in inflammation, along with an increase in the proliferation of keratinocytes, leading ultimately to wound contraction and re-epithelialization by the 14th day [78].

### 6.5. Salvia officinalis L.

#### 6.5.1. Ethnobotanical Uses and General Considerations

*S. officinalis* (Figure 1E), known as common sage or garden sage, is widely used as a seasoning and flavoring condiment in culinary arts. In traditional medicine, it is used for different ailments [24]. This species has, in fact, been the focus of careful scientific validation. Hence, numerous pharmacological activities have been reported such as anticancer, anti-inflammatory, antinociceptive, antioxidant, antimicrobial, antimutagenic, antidementia, hypoglycemic, and hypolipidemic [97]. The aerial parts have been used for wounds in the Iberian Peninsula, specifically through infusion used for baths and washes, to clean and enhance wound healing [14,61,66].

#### 6.5.2. Phytochemical Background

More than 120 different compounds have been identified in sage’s essential oils. Furthermore, this plant species presents chemical variations according to the parts used for extraction. Linalool is the main compound in stems, α-pinene and 1,8-cineole are predominant in the flowers, while bornyl acetate, camphene, camphor, humulene, limonene, and thujone are predominant in essential oils extracted from leaves (Figure 2). Besides, alcoholic and aqueous extracts have mainly flavonoids, such as luteolin-7-glucoside, while methanolic extracts present appreciable amounts of phenolic acids, such as caffeic and 3-caffeoylquinic acids. On the other hand, the infusion of *S. officinalis* has rosmarinic and ellagic acids as major bioactive compounds [36].

#### 6.5.3. Pharmacological Evidence

Karimbazeh et al. [79] evaluated the potential of the hydroethanolic leaf extract of *S. officinalis*. Firstly, a circular excision full-thickness wound was inflicted on the anterior-dorsal side of each rat to study the wound contraction ratio, period of re-epithelization, and histopathological change. Then, an incision wound was made through the skin and cutaneous muscle in the right side of depilated back. Ointments consisting of Eucerin^®^ (25%) and Vaseline^®^ (75%) with hydroethanolic extract of *S. officinalis* at 1, 3, and 5% were prepared and then topically applied once a day for 9 days. On the 10th day there was no sign of acute skin irritation in all tested animals. The highest tested concentration (5%) allowed an increase of the wound contraction and breaking strength ratio, and it also reduced the period of re-epithelialization. An increase in hydroxyproline content in dead space wounds was observed when compared to the control group, as it also promoted the formation of granulation tissue. Furthermore, *S. officinalis* proved to up-regulate macrophage and fibroblast distribution, increasing collagen deposition and promoting the proliferative stage of wound healing.

On the other hand, a study by Farahpour et al. [32] assessed the effect of *S. officinalis’* essential oil on infected wounds. After inoculation with *P. aeruginosa* and *S. aureus* and the infliction of circular full-thickness wounds, ointments containing 2 and 4% of *S. officinalis’* essential oil were applied once a day for 14 days. It showed a shortening of the inflammatory phase of the healing process once pro-inflammatory cytokines expression was reduced. Furthermore, cellular proliferation was stimulated via the upregulation of cyclin-D1 and Bcl-2 expression. By regulating FGF-2 and VEGF expressions, *S. officinalis* promotes neovascularization and tissue antioxidant status. In another study, Eshani et al. [98] used *S. officinalis* aqueous extract to synthesize ZnO/magnetite-based nanocomposites (ZnO/Mgt-NCs) that were further tested in both *in vitro* and *in vivo* experiments using infected wound models. Firstly, the *in vitro* assay demonstrated strong antibacterial properties when tested against *Streptococcus pyogenes* and *P. aeruginosa*. As for the *in vivo* assay, it revealed an improvement of the histological parameters and a decrease of the bacterial population growth on wounds treated with ZnO/Mgt-NCs. Additionally, granulation tissue, collagen density, and epithelization were improved. In turn, the development of a novel polyvinyl alcohol-based (PVA) nanofiber mat loaded with bioactive compounds from *Hypericum perforatum* L., *S. officinalis,* and *T. vulgaris* also revealed interesting antioxidant and antimicrobial activities, proving the potential of the innovative formulation [99].

### 6.6. Salvia rosmarinus Schleid. (Syn: Rosmarinus officinalis L.)

#### 6.6.1. Ethnobotanical Uses and General Considerations

According to the most recent phylogenetic criteria, the former scientific name *Rosmarinus officinalis* L. is considered a synonym of the currently accepted *Salvia rosmarinus* Schleid. (Figure 1F), since the genus *Rosmarinus* was merged into the genus *Salvia* [100]. Several ethnobotanical surveys report the healing effect of rosemary, either for wounds or burns, in the Portuguese traditional medicine [17,61,63,68]. Likewise, in Spain, Benítez et al. [19] reports the application of its essential oil to heal skin injuries, and Aceituno [65] states the vulnerary effect of a cataplasm made with leaves, while in Navarra this plant is traditionally used for the treatment of wounds, boils, and furuncles [14].

#### 6.6.2. Phytochemical Background

Since several chemotypes have been reported, most of the volatile compounds found in rosemary essential oil are monoterpenes, the most common ones being α-pinene, 1,8-cineole, borneol, limonene, and the ketones camphor and verbenone. Similarly, the sesquiterpene β-caryophyllene has been also identified [100,101,102]. Diterpenes such as rosmarol, carnosol, and carnosic acid, as well triterpenes such as ursolic and oleanolic acids have been found in appreciable amounts (Figure 2). The anti-inflammatory activity of rosemary is actually related to their synergic activity [100]. Phenolic acids such as rosmarinic and caffeic acids, and the flavonoids class (eriocitrin, luteolin 3′-*O*-β-D-glucuronide, hesperidin, diosmin, isoscutellarein 7-*O*-glucoside, hispidulin 7-*O*-glucoside, and genkwanin), comprise other important bioactive compounds found in the extracts of *S. rosmarinus* [100,102].

#### 6.6.3. Pharmacological Evidence

Once it is one of the topmost validated Lamiaceae species, a monograph about this medicinal plant can be found in the European Medicines Agency [103] and two other monographs are comprised in the European Pharmacopoeia [104]. *S. rosmarinus* finds important applications, namely in the food industry as a food additive and preservative [101], or as an active ingredient in cosmetic formulations [100].

Regarding the upcoming scientific evidence on the wound healing effect of *S. rosmarinus*, a recent clinical trial was conducted with 80 primiparous women to evaluate the healing effect of a rosemary cream on episiotomy. Results of this study indicated that on the 10th day postpartum, the group of women treated twice a day with rosemary cream had a statistically higher (*p* < 0.001) rate of episiotomy healing when compared to the placebo group [72].

The effects of S. *rosmarinus* on treating diabetic wounds have also been investigated. Abu-Al-Basal [10] studied its effect in alloxan-induced diabetic mice. After the infliction of full-thickness excision wounds, a group of mice was treated with *S. rosmarinus* essential oil (25 μL) twice a day for 3 days, while another group was submitted to an intraperitoneal injection of aqueous rosemary extract (10%) (0.2 mL), following an evaluation for 15 days. Statistically significant positive differences (*p* < 0.01) were found between rosemary-derived treatments and control groups. Furthermore, the essential oil treatment was more effective than the aqueous extract exhibiting an evident amelioration of several parameters related to the wound healing process. Similarly, Umasankar et al. [105] also confirmed the positive role of the essential oil of *S. rosmarinus* for wound closure of both streptozotocin-induced diabetic rats and non-diabetic rats. It is worth mentioning that a rosemary extract (5%) was also tested on full-thickness wounds inflicted on rabbits and the effect compared with povidone-iodine^®^ and isotonic saline solution [106].

In another interesting study, the wound healing effect of the essential oil of *S. rosmarinus* and *Melaleuca alternifolia* Cheel, in combination or separated, in chitosan-loaded formulations, were evaluated *in vivo* on excision wounds inflicted on rats. It was observed that the most successful chitosan-derived formulation was the one loaded with the combination of both essential oils, presenting a full re-epithelization with activated hair follicles, comparable to the positive control [82]. For instance, Khezri et al. [81] focused on encapsulated essential oil into nanostructured lipid carriers (EO-NLCs). Accordingly, gels with EO-NLCs and with only essential oil were prepared and applied to heal full-thickness wounds infected with *S. aureus* and *P. aeruginosa*. Results showed that EO-NLCs had antibacterial activity while promoting *in vivo* wound closure, angiogenesis, fibroblast infiltration, re-epithelialization, and collagen production. Moreover, an increase in key wound healing cytokines reduced inflammation and edema, while an increase in serum levels of VEGF may explain the observed positive angiogenic response [81]. Recently, Gavan et al. [107] aimed to develop carbomer-based hydrogel wound dressings containing ethanolic extracts of *S. rosmarinus* and two other plants (*Achillea millefolium* L. and *Calendula officinalis* L.). The *S. rosmarinus* loaded hydrogel was a promising formulation for wound healing therapy, similar to the hydrogel loaded with the mixture of the three studied extracts [107].

The healing effect of rosemary on burn wounds has also been tested by other authors [77,83,108,109]. Regarding the *in vivo* thermal-induced wounds, the activity of two different ointments prepared with *S. rosmarinus* and *Populus alba* L. essential oils was studied. Results at the end of 25 days of study showed *S. rosmarinus’* ointment to have a bigger wound contraction of 4.44 ± 0.07 cm^2^ compared to only 1.06 ± 0.44 cm^2^ for the Madecassol^®^ control group. Rosemary essential oil did not show acute toxicity and this report also suggests a significant healing activity during the proliferative phase of the wound healing process [83]. Recently, Khalil et al. [108] revealed the potential of an acetonic extract of *S. rosmarinus* against multidrug resistant pathogens that frequently infect burn wounds.

An Eucerin^®^-based cream containing rosemary’s essential oil at 2 and 4% was tested on excision wounds infected by *Candida albicans*. The 4% preparation yielded the best results compared to the other groups with a remarkable decrease in the infection and inflammation parameters. It also improved fibroblast proliferation, leading to complete wound contraction on the 16th day of the experiment [80]. The effect of Eucerin^®^-based formulations with rosemary extracts at different concentrations (15, 10 and 5%) was also evaluated on excision wounds of rats [110], as well as the effect of rosemary’s essential oil regarding the healing of skin lesions of mice [111]. In turn, one study evaluated in Wistar albino rats after plastic surgery reported the potential of rosemary’s essential oil to increase skin flaps survival, encompassing the avoidance of necrosis [112].

### 6.7. Salvia verbenaca L.

#### 6.7.1. Ethnobotanical Uses and General Considerations

*S. verbenaca* (Figure 1G) is a widespread *Salvia* species occurring in the Mediterranean region but also around Europe and in Asia [113]. This species exhibits various bioactive properties such as antibacterial, anticancer, antioxidant, antileishmanial, antidiabetic, immunomodulatory, and wound healing [113].

In Morocco, *S. verbenaca* is topically applied for burns healing and abscesses [114]. Regarding the Iberian traditional medicine, in Sierra Norte de Madrid (Spain), the leaves of *S. verbenaca* are macerated and prepared in olive oil as a vulnerary treatment for burns and wounds [65]. Benítez et al. [19] reported that in Western Granada (Spain), a decoction of the whole plant is used to treat skin injuries. Meanwhile, in the Ripollès district, Catalonia (Spain), the leaves are externally applied as an ointment to relieve pain and stimulate burns healing, thus functioning as an antipyrotic agent [15].

#### 6.7.2. Phytochemical Background

The chemical constitution of *S. verbenaca* has been analyzed from several plant organs, in both wild and cultivated plants, revealing mainly flavonoids, terpenoids, phenolic acids, alkaloids [113], and fatty acids [114]. The terpenoids identified in *S. verbenaca*’s essential oil are mostly α-pinene, β-pinene, sabinene, 1,8-cineole, β-phellandrene, linalool, *p*-cymene, linalyl acetate, (E)-β-ocimene, (Z)-β-ocimene, tricyclene, camphor, among many others. For instance, the phenolic acids identified in methanolic extracts were *p*-hydroxybenzoic acid, vanillic acid, rosmarinic acid, *p*-coumaric acid, caffeic acid, and phenolic diterpenes. Similarly, carnosol, carnosic acid, and methyl carnosate have also been identified (Figure 2) [113]. As for the main unsaturated fatty acids, linolenic and linoleic acids were assigned [114].

#### 6.7.3. Pharmacological Evidence

Guaoguao et al. [84] explored the effect of *S. verbenaca* extracts for the healing of second-degree burn injuries, in an SSD^®^ controlled study. It revealed that on the 9th day of the experiment, the skin injury areas were 29.17% for the cream base treatment, 44.34%, 47.55%, and 49.16% for the hexane, ethyl acetate, and *n*-butanol extracts, respectively, and 41.09% for the positive control SSD^®^. Results showed that the healing process was accelerated in rats submitted to the application of extract loaded creams. Noteworthy, the *n*-butanol extract was even more efficient than the standard SSD^®^ treatment. Following these experiments, Guaoguao et al. [115] assessed the safety of a hydroalcoholic extract obtained from the aerial parts of *S. verbenaca* that was successively fractionated with hexane, ethyl acetate, and *n*-butanol. Crude residues were massaged in the shaved rat’s healthy skin. After 14 days, animals were healthy and had no skin injuries, thus proving the absence of dermal toxicity or inflammation after the topical application of the extract.

In another interesting study, Righi et al. [29] evaluated the *in vivo* antiphlogistic potential of an hydromethanolic extract obtained from the aerial parts of this same *Salvia* species using a xylene-induced edema model. Mice treated with *S. verbenaca* extract reduced the weight increase caused by xylene, especially at 600 mg/kg of body weight that afforded 50% of edema inhibition. This concentration was as effective as the reference anti-inflammatory agents, such as indomethacin and dexamethasone, non-steroidal and steroidal anti-inflammatory drugs, respectively.

### 6.8. Thymus vulgaris L.

#### 6.8.1. Ethnobotanical Uses and General Considerations

*T. vulgaris* (Figure 1H), commonly known as thyme or garden thyme, is known for having an extensive array of pharmacological properties [6], also with important applications in culinary, perfumery, and cosmetics fields [116]. From this perspective, several pharmacological activities are reported, namely antibacterial, antioxidant, anti-inflammatory, antiviral, and anti-cancerous [116].

From the Iberian ethnomedicinal perspective of this thyme species, several reports have mentioned its potential application for wound healing. Firstly, a decoction of the aerial parts is referred to be used in the region of High River Ter Valley, Catalonia [64], as a vulnerary for the treatment of wounds. In Alt Empordà, Catalonia, Parada et al. [18] also documented the topical application of the aerial part of this species for the treatment of wounds. For instance, Cavero et al. [14] reported that in Navarra, the decoction of the aerial parts, is used to clean wounds and prepare ointments with wax, honey, and olive oil. The infusion of the aerial parts at the flowering stage is also reported to be used in the form of baths for wound healing [14,15].

#### 6.8.2. Phytochemical Background

In *T. vulgaris,* different classes of compounds can be found such as phenolics, terpenoids, flavonoids, steroids, alkaloids, tannins, and saponins. Phenolic compounds are the ones with more pharmacological significance, and among them, rosmarinic, caffeic, *p*-coumaric, geranic, *p*-hydroxybenzoic, gentisic, syringic, and ferulic acids have been highlighted. Thyme’s essential oil is mostly rich in thymol and carvacrol, but it also has geraniol, linalool, α- and β-pinene, *p*-cymene, and γ-terpinene (Figure 2). Apigenin, luteolin, cirsimaritin, genkwanin, and xanthomicrol are some of the identified flavonoids [116].

#### 6.8.3. Pharmacological Evidence

Regarding Mekkaoui et al. [77], *T. vulgaris* honey was mixed separately with the essential oils of O. vulgare, S. rosmarinus, and T. vulgaris, each at 0.5%. Interestingly, the honey containing *T. vulgaris’* essential oil provided the best outcomes in this study with wound closure rates for thermal and chemical-inflected burns of 85.21% and 82.14%, respectively. Furthermore, this treatment provided the shortest healing period of 14 days with a great potential to treat burn wounds compared to Madecassol^®^ or honey-alone treated animals. Notwithstanding, in a previous *in vivo* study, nitric oxide (NO), which was shown to be overproduced in thermal-induced wounds, progressively decreased during healing. However, in this investigation, thyme’s essential oil was shown to enhance the reduction of NO levels, presenting comparable results to conventional drugs used in burns management, such as SSD^®^. Rats treated with thyme’s essential oil also showed better results regarding the formation of new tissue [117].

In the Panah et al. [118] study, circular wounds were inflected on rats and after 21 days, results showed that the animals treated with ointments bearing thyme’s essential oil had a statistically significant better distribution of fibroblasts and macrophages, besides promoting angiogenesis and collagen deposition, when compared to the control group treated with Eucerin^®^ (25%) and Vaseline^®^ (75%). Similarly, the daily application of an ointment containing an ethanolic extract of *T. vulgaris* also afforded potent wound closure [119].

Given the substantial evidence showing the beneficial effects of T. vulgaris as a promising wound healing natural product, numerous studies have focused on developing innovative wound dressing systems. As an example, thyme’s essential oil was encapsulated in sodium caseinate (Na CAS) nanomicelles resulting in a gelatin nanocomposite hydrogel formulation. Interestingly, this study proved that this delivering system not only promotes wound closure, but also reduces the inflammatory factor IL-6, while encompassing an increase of VEGF and transforming growth factor-β1. Furthermore, an appreciable antibacterial activity was equally found with this hydrogel disrupting bacterial membranes followed by alkaline phosphatase leakage [31]. Another relevant study has led to the creation of an electrospun zein/thyme essential oil (TEO) nanofibrous membrane aiming to overcome some drawbacks of the conventional electrospun fibrous wound dressings, such as the lack of adjustment when topically applied on irregular wounds [33].

Chitosan films loaded with *T. vulgaris’* essential oil at 1.2% also showed suitable antioxidant activity given the high amounts of carvacrol, and antimicrobial activity against some bacterial strains such as *E. coli*, *Klebsiella pneumoniae, P. aeruginosa,* and gram positive *S. aureus* [25]. An antimicrobial wound dressing system of chitosan/Poly(ethylene oxide) nanofiber loaded with *T. vulgaris’* extract, synthetized through electrospinning, was equally afforded [120]. Similarly, the extract of *T. vulgaris* loaded into chitosan, eggshell membrane, and soluble eggshell membrane film were also attempted [121], and Poly(vinyl alcohol)-based nanofiber mats were also loaded with *T. vulgaris* hydroalcoholic extract [99].

## 7. Conclusions

Lamiaceae medicinal plants within the Iberian Peninsula, widely used by local communities of Portuguese and Spanish territories, comprise an important reservoir of ethnobotanical knowledge. This review highlights their significant potential for plant-based innovative wound healing therapies. According to the literature survey herein, the Lamiaceae species *L. stoechas*, *M. vulgare, O. vulgare*, *P. vulgaris*, *S. officinalis*, *S. rosmarinus*, *S. verbenaca,* and *T. vulgaris* gather a substantial amount of pharmacological evidence that attest their medicinal application and safety, and absence of side effects, which ultimately has already led to the design of some innovative drug delivery and wound dressing systems. We further suggest that investigating isolated compounds from these plants’ extracts should be explored and may serve as a prototype for even more effective therapeutics. Similarly, with this review we expect to inspire not only drug discovery and development researchers, but also clinicians to evolve additional and robust clinical trials that may attest how reliable these approaches can turn into the clinical practice. In this sense, we believe this review constitutes a paramount scientific basis to pave new avenues into wound management.

## Figures and Tables

**Figure 1 pharmaceuticals-16-00347-f001:**
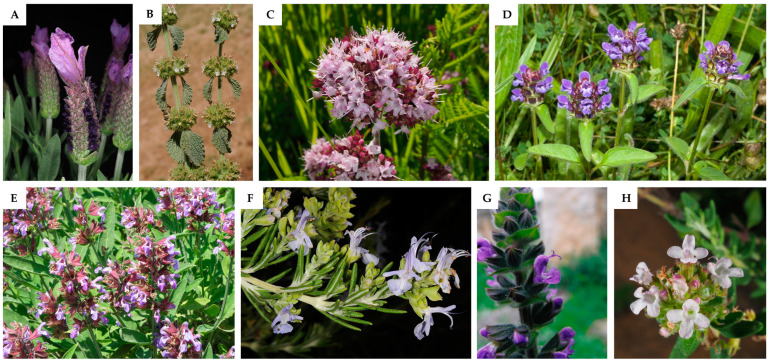
Lamiaceae species with scientific evidence for their wound healing use in the Iberian Peninsula. Images of the plants (**A**)—*Lavandula stoechas* L., (**B**)—*Marrubium vulgare* L., (**C**)—*Origanum vulgare* L., (**D**)—*Prunella vulgaris* L., (**E**)—*Salvia officinalis* L., (**F**)—*Salvia rosmarinus* Schleid (Syn: *Rosmarinus officinalis* L.), (**G**)—*Salvia verbenaca* L. and (**H**)—*Thymus vulgaris* L. were obtained from the website Plants of the World Online (http://www.plantsoftheworldonline.org/ accessed on 4 February 2023).

**Figure 2 pharmaceuticals-16-00347-f002:**
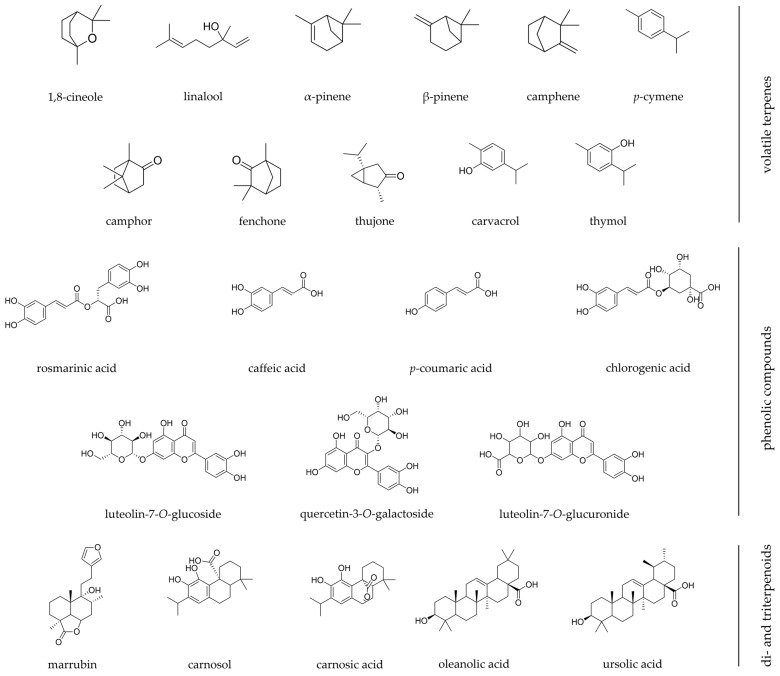
Chemical structures of the most relevant phytochemicals found among Iberian Lamiaceae species with wound healing scientific validation. Chemical structures were designed using ChemDraw software.

**Table 1 pharmaceuticals-16-00347-t001:** Lamiaceae species used in wound healing in the Iberian Peninsula.

Botanical Taxon	Popular Use	Part(s) Used	Method of Preparation and/or Use	Iberian Region	References
*Ajuga iva* (L.) Schreb.	Injuries	Aerial parts in flowering stage	Decoction	Western Granada, Spain	[19]
*Glechoma hederacea* L.	Wounds	Fresh or dried aerial parts; as fresh juice	Not reported	Trás-os-Montes, Portugal	[17]
*Lavandula lanata* Boiss.	Injuries	Essential oil	No preparation	Western Granada, Spain	[19]
*Lavandula stoechas* subsp. *luisieri* (Rozeira) Rozeira	Wounds	Dried flowers	Infusion and/or decoction; baths	Alentejo, Portugal	[61]
*Lavandula multifida* L.	Wounds	Aerial parts in flowering stage	Not reported	Trás-os-Montes, Portugal	[17]
*Lavandula pedunculata* (Mill.) Cav.	Injuries of the foot	Branches	Decoction	Arribes del Duero, Spain	[20]
*Lavandula stoechas* L. ^1^	Injuries	Aerial parts in flowering stage	Infusion	Western Granada, Spain	[19]
*Marrubium vulgare* L. ^1^	Injuries	Leaves of the stems not flowered	Cataplasms	Arribes del Duero, Spain	[20]
*Mentha suaveolens* Ehrh.	Wounds	Leaves	Decoction; baths	Penha Garcia, Portugal	[62]
	Wounds, cuts and dog bites	Aerial parts	Infusion; baths	Northwest of the Basque Country, Biscay and Alava, Spain	[67]
*Nepeta cataria* L.	Wounds	Flowering tops (from June to September) and whole plant (summer)	Not reported	Trás-os-Montes, Portugal	[17]
*Origanum vulgare* L. ^1^	Vulnerary	Flowers	Embrocation	Ripollès district, Pyrenees, Catalonia, Spain	[15]
*Phlomis purpurea* L.	Injuries	Aerial parts in flowering stage	Decoction	Western Granada, Spain	[19]
*Prunella vulgaris* L. ^1^	Wounds	Aerial parts and leaves	Smashed plant; cataplasm	Serra do Açor, Portugal	[60]
	Vulnerary	Aerial parts in flowering stage	Baths	Ripollès district, Pyrenees, Catalonia, Spain	[15]
	Wounds	Fresh leaves	Infusion	Protected Landscape of “Serra de Montejunto”, Portugal	[63]
*Salvia rosmarinus* Schleid. (Syn: *Rosmarinus officinalis* L.) ^1^	Burns and wounds	Not reported	Baths	Parque Natural de Montesinho, Portugal	[68]
	Wounds	Fresh or dried leaves	Infusion and/or decoction; baths	Alentejo, Portugal	[61]
	Wounds and ulcers	Aerial parts in flowering stage, and leaves	Not reported	Trás-os-Montes, Portugal	[17]
	Injuries	Essential oil	No preparation	Western Granada, Spain	[19]
	Wounds	Aerial parts	Boiled and placed between cloths. Clean with decoction; crush with a little white wine; ointment with wax, olive oil and a small glass of red wine.	Navarra, Spain	[14]
	Wounds and burns	Leaves	Cataplasms made of leaves and butter or olive oil	Sierra Norte de Madrid, Spain	[65]
*Salvia blancoana* subsp. *vellerea* (Cuatrec.) W.Lippert	Injuries	Aerial parts in flowering stage	Infusion or decoction	Western Granada, Spain	[19]
	Injuries	Essential oil	No preparation	Western Granada, Spain	[19]
*Salvia officinalis* L. ^1^	Wounds	Aerial parts	Baths	Alentejo, Portugal	[61]
	Wounds	Aerial parts	Infusion; baths	Riverside, South Navarra, Spain	[66]
	Wounds	Aerial parts	Clean with infusion	Navarra, Spain	[14]
*Salvia sclarea* L.	Wounds and burns	Leaves and roots	Cataplasms made of heated leaves, olive oil and lard. Roots’ decoction for baths	Parque Natural de Montesinho, Portugal	[68]
*Salvia verbenaca* L. ^1^	Burns and wounds	Leaves	Maceration with olive oil	Sierra Norte de Madrid, Spain	[65]
	Injuries	Whole plant	Decoction	Western Granada, Spain	[19]
	Antipyrotic	Leaves	Ointment	Ripollès district, Pyrenees, Catalonia, Spain	[15]
*Clinopodium nepeta* subsp. *spruneri* (Boiss.) Bartolucci & F.Conti	Wounds	Flowering tops	Not reported	Trás-os-Montes, Portugal	[17]
*Satureja hortensis* L.	Wounds	Flowering tops	Not reported	Trás-os-Montes, Portugal	[17]
*Sideritis hirsuta* L.	Injuries	Aerial parts in flowering stage	Decoction	Western Granada, Spain	[19]
*Sideritis incana* L.	Injuries	Aerial parts in flowering stage	Decoction	Western Granada, Spain	[19]
*Teucrium scorodonia* L.	Wounds and cuts	Aerial parts	Decoction; baths and cataplasms	Northwest of the Basque Country, Biscay and Alava, Spain	[67]
	Infected wounds	Aerial parts	Cataplasms	Northwest of the Basque Country, Biscay and Alava, Spain	[67]
*Thymus mastichina* (L.) L.	Injuries	Aerial parts in flowering stage	Decoction	Western Granada, Spain	[19]
	Injuries	Essential oil	No preparation	Western Granada, Spain	[19]
*Thymus pulegioides* L.	Wounds	Aerial parts in flowering stage	Lotions and cataplasms	Trás-os-Montes, Portugal	[17]
*Thymus vulgaris* L. ^1^	Wounds	Aerial parts in flowering stage	Decoction	High river Ter Valley, Catalonia, Spain	[64]
	Wounds	Not reported	Topical	Alt Empordà, Catalonia, Spain	[18]
	Wounds	Aerial parts	Clean with decoction; ointment with wax, honey, and olive oil	Navarra, Spain	[14]
	Wounds	Aerial parts in flowering stage	Clean with infusion	Navarra, Spain	[14]
	Vulnerary	Aerial parts in flowering stage	Baths	Ripollès district, Pyrenees, Catalonia, Spain	[15]
*Thymus zygis* L.	Injuries of the foot	Branches	Decoction made with branches of *T. mastichina* and *L. pedunculata*	Arribes del Duero, Spain	[20]
*Thymus zygis* subsp. *gracilis* (Boiss.) R.Morales	Injuries	Essential oil	No preparation	Western Granada, Spain	[19]

^1^ These species have pharmacological validation that is further discussed in Section 6.

**Table 2 pharmaceuticals-16-00347-t002:** Clinical trials using Iberian Lamiaceae species with wound healing activity.

Plant Name	Plant Parts	Type of Extraction	Formulation	Type of Wound	Evaluation Period	References
*Lavandula stoechas* L.	Inflorescences	Hydrodestillation	Essential oil at 1.5% was incorporated in olive oil as carrier and 5 to 7 drops of the oil was added to 4 L of water in sitz baths	Episiotomy	10 days	[69]
		Hydrodestillation	Herbal cream containing *L*. *stoechas* and *Pelargonium roseum* Ehrh. essential oils and *Aloe vera* gel	Second-degree skin burns	14 days	[70]
*Origanum vulgare* L.		Water extraction	*O. vulgare* water extract at 3% was incorporated in an ointment	Excision wounds	90 days	[71]
*Salvia rosmarinus* Schleid (Syn: *Rosmarinus officinalis* L.)		Ethanol (70%) extraction	The hydroalcoholic extract at 3% was incorporated in Farabi^®^ base cream	Episiotomy	10 days	[72]

**Table 3 pharmaceuticals-16-00347-t003:** Key findings of in vitro and *in vivo* studies using Iberian Lamiaceae species with wound healing scientific validation.

Plant Name	Plant Parts	Type of Extraction	Formulation	Type of Study	Model of Study	Sample Type of Wound	Evaluation Period	References
*Lavandula stoechas* L.	Leaves	Residual water from hydrodistillation		*In vitro*	Human fibroblasts (HFF1)	Scratch wound healing assay	0, 24, 48, 72, and 96 h	[73]
	Leaves, stems, and inflorescences	Hydrodestillation	Essential oil at 0.5% was incorporated into a topical cream emulsion	*In vivo*	Male Wistar rats	Excision wounds	16 days	[74]
	Aerial parts	Methanol extraction	Methanolic extract was incorporated in petroleum jelly at 5 and 10%	*In vivo*	Albino Wistar rats	Excision wounds	18 days	[8]
*Marrubium vulgare* L.	Leaves	Methanol (80%) extraction		*In vitro*	Normal Human Dermal Fibroblasts (NHDF)	A gap was established between seeded cells	48 h	[7]
	Aerial parts	Acetone extraction	Acetonic extract at 5% was incorporated in an ointment of glycol stearate, propylene glycol, and liquid paraffin (3:6:1)	*In vivo*	Rats	Excision wounds	15 days	[23]
	Leaves	Hydroethanol extraction	The hydroethanolic extracts from *M. vulgare*, *D. viscosa,* and their mixture were incorporated in Vaseline^®^	*In vivo*	Male Wistar rats and male mice	Burn wounds	21 days	[3]
*Origanum vulgare* L.	Leaves	Water extraction	The extract was included in green synthesized titanium dioxide nanoparticles (TiO_2_.NPs)	*In vivo*	Wistar male Albino rats	Full thickness excision wounds	12 days	[75]
	Aerial parts	Hydrodestillation		*In vitro*	Human keratinocytes (NCTC 2544)	Scratch wound healing assay	72 h	[26]
		Hydrodestillation	The essential oil at 5% was encapsulated in Poly (L-lactide-co-caprolactone) (PLCL) and silk fibroin (SF) through electrospinning	*In vivo*	Male Sprague-Dawley rats	Silicone-splinted full-thickness cutaneous wounds	10 days	[30]
		Hydrodestillation	An emulsion of essential oil at 10% and zinc oxide nanoparticles were both included in electrospun nanofibrous membranes	*In vivo*	Male Sprague-Dawley rats	Streptozocin-induced diabetic wounds	15 days	[76]
		Hydrodestillation	Essential oils of *T. vulgaris*, *O. vulgare* and *R. officinalis* at 0.5% were incorporated in 100 g of *T. vulgaris* honey	*In vivo*	Albino Rabbits	Chemical and thermal wounds	2 weeks	[77]
*Prunella vulgaris* L.	Whole plant	Water extraction	Aqueous extract at 10% was incorporated in a cream	*In vivo*	Female albino Sprague-Dawley rats	Partial thickness burns	14 days	[78]
	Flowering aerial parts	Methanol (80%) extraction followed by fractionation in *n*-hexane, ethyl-acetate, and water	Extracts at 1% were incorporated in ointments	*In vivo*	Male Wistar albino rats and mice	Linear incision and circular excision wounds	12 days	[27]
*Salvia officinalis* L.	Leaves	Hydroethanol extraction	Hydroethanolic leaf extract at 1, 3, and 5% was incorporated in an Eucerin^®^- and Vaseline^®^-based formulation	*In vivo*	Male Wistar rats	Full-thickness excision and incision wounds	9 days	[79]
	Leaves	Hydrodestillation	Essential oil at 2 and 4% with yellowsoft paraffin	*In vivo*	Male BALB/c mice	Full-thickness wounds	14 days	[32]
*Salvia rosmarinus* Schleid (Syn: *Rosmarinus officinalis* L.)	Aerial parts	Hydrodestillation and water extraction		*In vivo*	Male BALB/c mice	Full-thickness excision wounds	15 days	[10]
	Leaves	Hydrodestillation	Essential oil at 2 and 4% was incorporated in an Eucerin^®^-based cream	*In vivo*	Male Wistar rats	Excision wounds	20 days	[80]
		Hydrodestillation	Essential oil alone and nanostructured lipid carriers loaded with essential oil were incorporated in gel formulations	*In vivo*	Mice	Circular full-thickness wounds	14 days	[81]
		Hydrodestillation	Chitosan-based formulation loaded with essential oils of *S. rosmarinus* and *M. alternifolia* at 10%	*In vivo*	Male Sprague-Dawley rats	Full-thickness excision wounds	14 days	[82]
	Aerial parts	Hydrodestillation	Essential oils of *S. rosmarinus* and *P. alba* were incorporated in ointments	*In vivo*	Male Wistar rats	Second-degree burn wounds	25 days	[83]
*Salvia verbenaca* L.		Hexane, ethyl acetate, n-butanol extractions	Hexane, ethyl acetate and n-butanol were incorporated in creams	*In vivo*	Rats	Second-degree burn wounds	9 days	[84]
*Thymus vulgaris* L.		Hydrodestillation	*T. vulgaris* honey separately mixed with the essential oils of *O. vulgare*, *S. rosmarinus*, and *T. vulgaris*, each one at 0.5%	*In vivo*	Rabbits	Thermal and chemical burn wounds	14 days	[77]
	Leaves	Hydrodestillation	*T. vulgaris* essential oil encapsulated in sodium caseinate nanomicelles	*In vivo*	Albino male rats	Full-thickness skin wounds	18 days	[31]

## Data Availability

Not applicable.

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
