# Peer review of "Exploring Iberian Peninsula Lamiaceae as Potential Therapeutic Approaches in Wound Healing"

_pharmaceuticals, 2023, doi:10.3390/ph16030347_

Round 1

Reviewer 1 Report

Dear Authors,

Please focus on the points below,

1. Introduction part elaborated. Kindly make it succint.

2. There are grammer and spelling mistakes. Kindly check and correct them.

3. Athough medical plants described have many beneficial effects over synthesized medicinal drugs. What about time required for wound healing with either of the medicines, how do you compare and justify? How these are helpful in emergency cases? 

Author Response

Response to reviewers’ comments (author responses are noted in italic)

We thank the reviewers for their time reading our manuscript and for their valuable comments and suggestions. We are grateful for these comments and the article improved as a result. Therefore, we hope to have answered all queries noted, as part of the revision process for the manuscript, which were addressed either in the manuscript or in the rebuttal letter below, as relevant.

Comments from Reviewer 1

We thank the referee for the valuable comments and suggestions, and we have responded to each comment below.

Introduction part elaborated. Kindly make it succinct.

Thank you for your valuable observation. The authors reviewed the Introduction section and agreed to make it clearer (some text (lines 54 to 58; 68 to 72; 84 to 87) was removed).

There are grammar and spelling mistakes. Kindly check and correct them.

Thank you for your comment. The text was carefully revised by the authors to check grammar and spelling mistakes that were properly corrected.

Although medical plants described have many beneficial effects over synthesized medicinal drugs. What about time required for wound healing with either of the medicines, how do you compare and justify? How these are helpful in emergency cases? 

Thank you for raising these helpful questions. As it is discussed throughout the manuscript, for each Lamiaceae species with wound healing activity, several authors have reported that such natural products frequently exhibit better results when compared to the effect of conventional drugs (such as madecassol®, povidone-iodine® and silver sulfadiazine®) currently used in therapeutics. Noteworthy, an improvement concerning the time required for wound healing by applying these natural products is also mentioned in several reports and, therefore, presented as well in the manuscript (lines 311 to 315; 322 to 325; 331 to 334; 511 to 513; 620 to 623; 650 to 653; 746 to 748).

Reviewer 2 Report

The review presented by Marques et al., is well presented and offers an interesting premise. Below I express my comments and suggestions that I believe, may improve the manuscript in its current form:

Major

I found the lack of Figures and schemes a noteworthy issue. Please consider adding some; for instance to illustrate skin homeostasis or something akin. Also, chemical structures of the natural products discussed in text would be appreciated.

Also, I believe that a paragraph introducing the medicinal role of the Labiatae family could benefit the manuscript. In particular, the antibacterial potential can be the major focus due to its relation towards skin healing

A suggestion for table 3 would be to mention if the composition of extract is known or determined.

Minor comments:

Line 15: Change barrier to tissue

Line 16: Remove the word skin

Line 36: Change "and acts as an immune defence against microorganisms " to "plus it acts as an [...]"

Line 45: Supress the additional spacing between undergo and normal

Line 65: I'm not sure if defended is the proper term to use in this context, my suggestion is to change it.

Line 69: Idem

Author Response

Response to reviewers’ comments (author responses are noted in italic)

We thank the reviewers for their time reading our manuscript and for their valuable comments and suggestions. We are grateful for these comments and the article improved as a result. Therefore, we hope to have answered all queries noted, as part of the revision process for the manuscript, which were addressed either in the manuscript or in the rebuttal letter below, as relevant.

Comments from Reviewer 2

We thank the referee for the valuable comments and suggestions, and we have responded to each comment below.

I found the lack of Figures and schemes a noteworthy issue. Please consider adding some; for instance, to illustrate skin homeostasis or something akin. Also, chemical structures of the natural products discussed in text would be appreciated.

Thank you for your valuable suggestion. The authors considered the importance of introducing figures and provided two in the present resubmitted manuscript. Figure 1. presents the eight species discussed for their wound healing evidence, and Figure 2. illustrates the chemical structures of the most relevant compounds found in the plant species discussed for their wound healing activity.

Also, I believe that a paragraph introducing the medicinal role of the Labiatae family could benefit the manuscript. In particular, the antibacterial potential can be the major focus due to its relation towards skin healing.

Thank you for your suggestion. The authors introduced a section in Introduction, about the relevant role of some Lamiaceae-related bioactivities, with a special focus on antibacterial, and how they positively influence wound healing (line 70 to 76).

A suggestion for table 3 would be to mention if the composition of extract is known or determined.

Thank you for your comment. The authors recognize your concern. However, adding another column would make Table 3 more exhaustive and difficult to interpret. We aim to make it succinct and straightforward. Besides, phytochemical information about each pharmacological validated species is also provided in each specific section within the text.

Line 15: Change barrier to tissue.

Thank you for your comment. The word was modified accordingly.

Line 16: Remove the word skin.

Thank you for your comment. The word was removed accordingly.

Line 36: Change "and acts as an immune defense against microorganisms " to "plus it acts as an [...]".

Thank you for your comment. The text was modified accordingly.

Line 45: Suppress the additional spacing between undergo and normal.

Thank you for your comment. The space was suppressed as suggested.

Line 65: I'm not sure if defended is the proper term to use in this context, my suggestion is to change it.

Thank you for your comment. The term “defended” was substituted by “suggested”.

Line 69: Idem

Thank you for your comment. To address a suggestion made from Reviewer 1, this part within the Introduction section has been removed as well as the term “defended”.

Round 2

Reviewer 2 Report

The authors have made substantial improvements to the manuscript. Most of my suggestions and concerns have been addressed.

However, I still found some minor errors on some wording or style in the main text.

Also, I noticed that the author list has changed. This by itself is not unusual; still, I wonder which contributions were made by him.

Author Response

Dear Reviewer,

Thank you for your comments.

Regarding the changing in authorship, it is because at the moment of submission is the moment when I add the authors names and affiliations, and because we had some other papers being submitted close to each other, I was confused and it was my fault that I did not considered the name of Paulo Oliveira. His contribution was as I mentioned in the document attached to the revision, that was review & editing, and also supervision, because we are both supervisors of the 1st author.